# BTS: Building Timeseries Dataset:
# Empowering Large-Scale Building Analytics

**Arian Prabowo**[1], **Xiachong Lin**[1], **Imran Razzak**[1], **Hao Xue**[1], **Emily W. Yap**[2], **Matthew Amos**[3], **Flora D. Salim**[1]

[1]CSE, UNSW *, Sydney NSW 2052
[2]SBRC †, UOW, Wollongong NSW 2522
[3]Energy, CSIRO ‡, Newcastle NSW 2304
{arian.prabowo, imran.razzak, hao.xue1, flora.salim}@unsw.edu.au
dawn.lin@student.unsw.edu.au    eyap@uow.edu.au    matt.amos@csiro.au

## Abstract

Buildings play a crucial role in human well-being, influencing occupant comfort, health, and safety. Additionally, they contribute significantly to global energy consumption, accounting for one-third of total energy usage, and carbon emissions. Optimizing building performance presents a vital opportunity to combat climate change and promote human flourishing. However, research in building analytics has been hampered by the lack of accessible, available, and comprehensive real-world datasets on multiple building operations. In this paper, we introduce the Building TimeSeries (BTS) dataset. Our dataset covers three buildings over a three-year period, comprising more than ten thousand timeseries data points with hundreds of unique classes. Moreover, the metadata is standardized using the Brick schema. To demonstrate the utility of this dataset, we performed benchmarks on the multi-label timeseries classification task. This task represent an essential initial step in addressing challenges related to interoperability in building analytics. Access to the dataset and the code used for benchmarking are available here: https://github.com/cruiseresearchgroup/DIEF_BTS

## 1 Introduction

**Importance of building analytics.** Building analytics, also known as data-driven smart building [12], involves the automated adjustment of building operations to minimize emissions and costs, optimize energy usage, and enhance indoor environmental quality and occupant experience, including comfort, health, and safety [72]. This is particularly crucial given that buildings account for a third of global energy usage and a quarter of global carbon emissions, comparable to the transport sector [27]. Optimizing building performance has the potential to significantly mitigate climate change and promote human well-being.

**Literature gaps.** This paper addresses two critical gaps in building analytics research. Firstly, in Section 2.1, we highlight the scarcity of publicly available and freely accessible datasets on comprehensive real-world building operations, as exemplified in Table 1. While LBNL59 [49, 36] is the only dataset that captures various aspects of building operations comprehensively, it only includes data from a single building.

---

*School of Computer Science and Engineering (CSE), University of New South Wales (UNSW).
†Sustainable Buildings Research Centre (SBRC), University of Wollongong (UOW).
‡Commonwealth Scientific and Industrial Research Organisation (CSIRO).

38th Conference on Neural Information Processing Systems (NeurIPS 2024) Track on Datasets and Benchmarks.

Table 1: **Comparing the scope of representative datasets for building analytics**. Only datasets on real-world building operations that are available, accessible are presented. Note that non-intrusive load monitoring (NILM) is not a single dataset but a task that usually use similar datasets. Similarly, AshraeOB is also a collection of dataset.

| Year | Dataset | Unique Class | Scope |
|------|---------|--------------|-------|
| 2013 | SLRHOME [5] | 3 | Aggregate energy load and generation |
| 2014 | LCLD [79] | 2 | Aggregate energy load and tarriff |
| 2015 | UCI [78] | 1 | Aggregate energy load |
| 2017 | BGD2 [51] | 18 | Detailed energy load |
| 2020 | LBNL59 [36, 49] | 35 | **Comprehensive** |
| 2021 | AshraeOB [18, 47] | 76 | Occupancy and their factors (e.g. lighting, indoor climate) |
| Varies | NILM [72] | Varies | Detailed energy load |
| 2024 | **BTS (Ours)** | 215 | **Comprehensive** |

This limitation underscores the need for datasets covering multiple buildings to address the second gap: interoperability in building analytical models. Interoperability is crucial for scalability, allowing models to be applied across diverse buildings with differing characteristics such as climate, usage, size, regulations, budget, and architecture. This challenge is discussed further in Section 2.2. Additionally, such datasets inherently possess properties of interest to machine learning research, such as domain shift, multimodality, imbalance, and long-tailedness, which are discussed further in Section 2.3.

**Building TimeSeries (BTS): A new dataset.** In this paper, we introduce a new anonymized building analytics dataset sourced from three buildings located in undisclosed regions across Australia. Spanning a three-year period, our dataset encompasses over ten thousand timeseries data points, featuring a diverse array of 240 unique classes. Notably, this surpasses the ontological breadth of LBNL59 by more than threefold. These ontologies serve as standardized categorizations of building timeseries data, including parameters like `Temperature_Setpoint` and `Voltage_Sensor`. The breadth of ontologies within our dataset enables researchers to explore buildings with more intricate analytics setups, facilitating deeper insights into building dynamics and performance. Furthermore, the meta-data are standardized using the popular Brick schema [7], ensuring consistency and compatibility across analyses.

**A Benchmark.** To demonstrate the utility of this dataset, we conducted benchmarks on a machine learning model interoperability task: multi-label timeseries classification. One of the initial steps in achieving building analytics interoperability is to map thousands of heterogeneous timeseries generated from sensors and actuators to a standardized ontology, such as the Brick schema [7]. This is also known as the timeseries ontology classification task [67].

We also performed an additional benchmark on a zero-shot forecasting task [19, 28]. This explores scenarios where a building manager deploys a pre-trained model without fine-tuning. This task is more complex than typical setups because the model must generalize to an arbitrary number of timeseries, various permutations of their ontologies, and their relationships [45]. The details can be found in Appendix D.

**Contribution.** This paper introduces the Building TimeSeries (BTS) dataset, addressing critical gaps in publicly available building analytics datasets. Existing datasets often lack accessible, available, comprehensive, real-world, building operations data, hindering progress in building analytics research. While some datasets like LBNL59 offer a holistic view, they are limited to single buildings, impeding efforts to achieve interoperability in building analytics models. BTS fills this void by providing data from three diverse buildings, spanning a three-year period and encompassing over ten thousand timeseries data points and 240 unique classes. Morever, BTS inherently possess properties relevant to machine learning research, including domain shift, multimodality, imbalance, and long-tailedness. Furthermore, we conduct a benchmark on a machine learning model interoperability task — multi-label timeseries classification — demonstrating BTS's utility in addressing challenges related to interoperability in building analytics. Overall, BTS dataset advances the pursuit of optimizing building performance, ultimately aiding efforts to mitigate climate change and enhance human flourishing.

## 2 Related Works

### 2.1 Existing Datasets

To write this section, we reviewed of the building datasets utilized in the literature. We found that, in most cases, the datasets are private, static, simulation-based, or limited in ontology. Although our review is not systematic as this is not a review paper, our search was sufficiently extensive to ensure the validity of our findings. The datasets discussed here are primarily derived from five recent review papers [59, 72, 39, 40, 44] along with our own collections. This would have included earlier surveys such as [6]. Table 4 in the appendix list the works mentioned in this section.

**Availability and Accessibility**. Most research on building analytics uses private datasets [82]. This is due to security and privacy concerns of building owners and occupants. This is prevalent across many aspects of building analytics, from HVAC [69, 33, 77, 71, 30, 29, 20], energy use [60, 61], and more holistic systems [34, 35, 23, 42, 43, 67].

Some datasets are publicly accessible, but not for free, such as Pecan Street [14], or not freely available, such as ecobee [21]. Notably, the Mortar dataset [22], which comprises data from 90 buildings and over 9.1 billion data points, is currently unavailable due to cloud deployment issues at the time of writing.

**Building Operation**. Most public datasets such as EUBUCCO [53] only contain static information such as type, height, and construction year. However, these datasets do not contain sufficient information on building operation. Others contain more extensive information, such as PLUTO [17] and GBMI [10] with more than 70 fields and 380 fields respectively, or building polygons [87] and 3D shapes [9].

While many public datasets include time information, they are often too sparse (yearly) to be useful for building analytics, which require at least daily data. Examples include the popular CBECS [16], and larger ones like BERTOOL [75] and CENED+2 [68], each containing about a million instances.

**Real-World and Not Simulation**. Simulations, while valuable, present limitations due to their reliance on assumptions that may not accurately reflect real-world building systems and human behaviors [95, 72]. Results have been shown to diverge from actual telemetry data in multiple studies [74, 1, 76]. These simulations are often calibrated to match existing datasets such as BEM4CBECS [2, 91, 92, 90] which are based on the CBECS dataset [16], while ResStock [84] and ComStock [58] are based on data from 2.3 million meters in the US [85]. Another notable examples are CityLearn Challenge Series [81, 54, 57, 56]. Not all simulations are software-based. There are also hardware-in-the-loop laboratory setup [65, 64].

**Whole Building Scope**. The few remaining datasets are listed on Tab. 1. They have limited scope, and does not fully capture the entire building as a holistic system. For example, most datasets are focused only on aggregated energy load (UCI [78]), or disaggregated (ASHRAE [32, 31, 37], BDG [52, 51], NILM [59]), or when combined with generation [5], or price [79]. Others focuses on occupancy patterns [25, 24, 18, 47] or water [13, 70].

To our knowledge, LBNL59 [49, 36], a medium-sized office building in Berkeley, is the only comprehensive existing dataset. Our dataset complements this dataset by introducing three new buildings, with more diverse ontology. This allows the exploration various transfer learning techniques to ensure that machine learning models are interoperable between buildings. In Section 3.2, we make a detailed comparison of LBNL59 with our dataset.

### 2.2 Relevant Challenges in Building Analytics

The standardization of building timeseries data overcomes the challenge of interoperability and scalability that can give rise to greater widespread adoption of energy flexibility in a systematic manner. Achieving zero-energy buildings has two conflicting optimization goals: to maximise occupant comfort and indoor environmental quality, and to minimise carbon emissions and operating costs [41]. It involves two components: the building model that represents the thermodynamics and energy behavior of a building and its components such as its construction, materials, and HVAC system, and secondly, a control strategy to automate the control operations.

Obtaining a building model involves expert knowledge and significant time to develop and validate. This is further amplified by requiring individual models for each building. These models can be white-box (physics-based) [95, 80], black-box (data-driven), or grey-box (hybrid) [50, 46]. Our dataset and benchmark experiment, which automate timeseries data classification, help address this challenge by reducing the time and cost associated with building key components of these models.

In comparison to building models, there has been a significant focus on optimising building control operations and transitioning from conventional rule-based approaches to model predictive control or data-driven methods [50]. The Building Optimization Framework or BOPTEST [11] exists to enable the development and benchmarking of building control strategies. The performance of a control strategy or algorithm is evaluated on a virtual "test case". Currently, these test cases are simulation physics-based models of ideal buildings developed on Spawn [83] (a co-simulation of Modelica and EnergyPlus) and act as emulators. In their paper, Blum et al. [11] make the contrasting argument that simulation-based test cases offer advantages over existing challenges when testing in real buildings, such as being time-consuming and subject to stochastic events.

However, accessing publicly available and anonymized building timeseries data from various non-residential building types acts as a commodity to reduce the time to develop individual hybrid building models. On one hand, using data from real buildings can be used to calibrate and interpolate lesser-known parameters, while maintaining moderate interpretability. And on the other hand, using standardized timeseries data such as the datasets introduced here aids in scalability and deployability to build generalized multi-zone environments and substituting with data from another building system or zone.

More broadly, there are various other applications of this dataset. **Generative AI for Privacy-Preserving Data Sharing**: Explore the use of generative AI to create synthetic building timeseries data, enabling building owners to contribute data for research while safeguarding sensitive information. **LLM Integration for Natural Language Interaction**: Investigate methods to integrate LLMs with building timeseries data, allowing various stakeholders such as building operators to interact with and query the data using natural language. **Redeployability**: By using a standardised ontology to describe the building, and linking timeseries data to the building model, applications (e.g. measurement and verification, chiller scheduling, occupant comfort) can be written to deploy against a fleet of buildings without a deep understanding of the building topology, such as those provided within this dataset.

### 2.3 Relevant Challenges in Machine Learning (ML) Research

**Domain shift and domain adaptation.** In the realm of ML research, one challenge is in domain adaptation, particularly about the diverse characteristics of buildings. These variations encompass factors such as climate, usage, size, regulations, budget, and architecture, resulting in notable distribution shifts. Consequently, traditional ML methodologies fall short in address these discrepancies. Therefore, the development and implementation of domain adaptation techniques [4, 3, 73, 26] are crucial to ensure model generalization across different buildings. Additionally, the usual alternative of employing large foundational models [94] is impractical because privacy and security concerns limit the availability of extensive building datasets for training. Moreover, as shown in Section 3.3.2, the unique permutation of ontologies in each building further complicates the scenario, necessitating novel approaches capable of handling arbitrary permutations effectively [45]. This is an issue since many timeseries architecture do not allow the model to input and output an arbitrary number of variate [86].

**Multimodal Learning with knowledge graphs (KG) and unbalanced multivariate timeseries (MVTS) with long tails.** While many studies focus on MVTS data in conjunction with spatial graph [62, 63], video, image, audio, and text data [15, 89], research on MVTS with knowledge graphs is scarce. Our dataset enable such research as it contains the Brick schema which is a KG on building metadata, describing relationship between the timeseries in the MVTS. Our dataset is also challenging because it is unbalanced and featuring distributions long tails. As shown in Section 3.3.2, some classes, like `Chilled Water Differential Temperature Sensor`, might only have one or two instances in the entire dataset, or, like `Alarm`, have zero values for most of the time . These challenges could fuel the developments of innovative techniques.

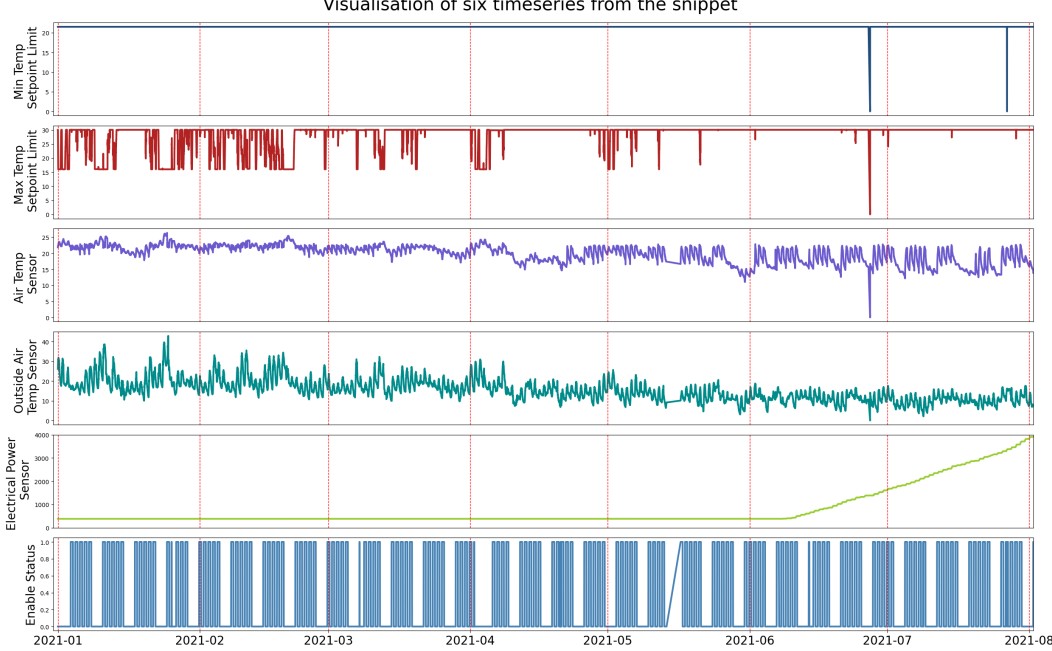

Figure 1: Visualisation of six timeseries with varying classes. The data is from the snippet of our BTS dataset available at `https://github.com/cruiseresearchgroup/DIEF_BTS`

## 3 Dataset

### 3.1 Collection Process

This dataset is comprised of data collected onto CSIRO's Data Clearing House (DCH `https://research.csiro.au/dch/`) digital platform [38]. Connecting to the Building Management Systems (BMS), timeseries data is collected from sensors, power, water and gas meters, and other devices within the buildings and uploaded using Message Queuing Telemetry Transport Secured (MQTTS). A semantic model of the building was created using DCH platform tooling. This created Brick schema [7] class definitions (version 1.2.1) for points within the model, and linked these points to the timeseries data ingested via MQTTS.

All instrumentation was conducted prior to the study, and as such no equipment installation or hardware setup was required by the authors. The work integrates with DCH platform which provides digital infrastructure to house building data, as well as to generate semantic models to describe the topology and instrumentation installed within the building. Based on a previously conducted systemic evaluation of existing ontologies suitable for our research context, we chose the Brick schema [66]. In terms of effort to map to the Brick schema, once sufficient details about the building are compiled, then typically expert engineers requires at least one to two days of per building to generate a full semantic building model.

Identifiers for both the point within the model, and the timeseries identifier were anonymised by generating Universally Unique Identifiers (UUID), and a three-year-period subset of the timeseries data was extracted from the DCH platform to produce this dataset. The data was not cleaned in effort to allow evaluation of various different cleaning algorithm, and to allow the evaluations of algorithms against data with realistic errors.

### 3.2 Description

**The Building TimeSeries (BTS) dataset** provides comprehensive, real-world data on building operations from three buildings in undisclosed Australian locations. It includes timeseries data (visualized in Figure 1) and building metadata standardised according to Brick schema [7]. Table 2

shows the statistics, comparing it to the LBNL59 dataset which is the only comparable dataset currently available. Part of this dataset have been presented in [45].

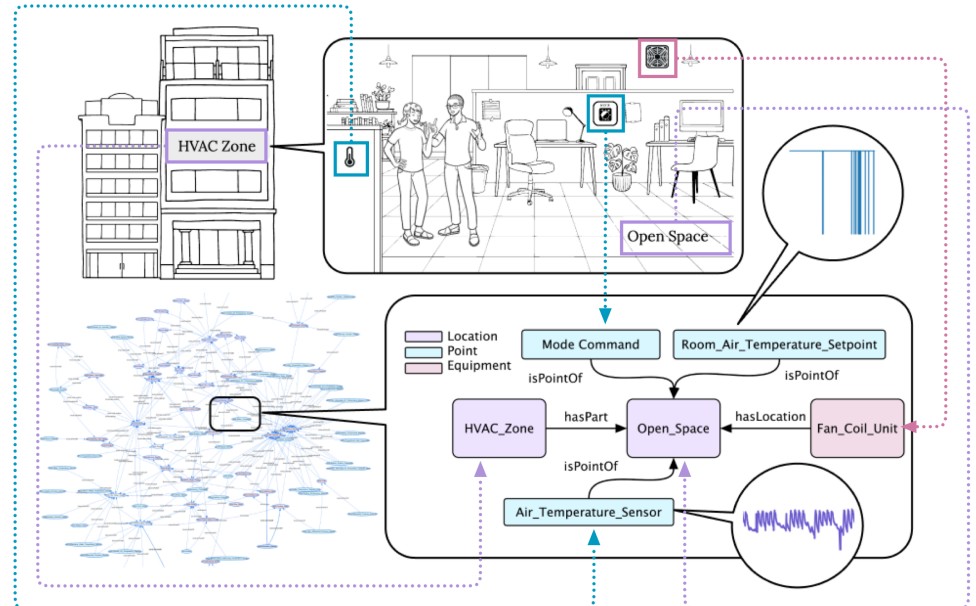

Figure 2: **Brick Schema Illustration and Visualization,** depicting machine-readable metadata for buildings as a knowledge graph. It reveals the logical and spatial links between distinct entities within a building, including the associated timeseries.

**Our dataset use the Brick schema**, a knowledge graph (KG) that details building components and their logical and spatial relationships. As illustrated in Figure 2, it specifies the equipment present in the buildings, the sensors attached to these equipment, their locations, and other related components within the same vicinity. Moreover, it also standardised the categorisations of the timeseries data into classes. The formal definition of the KG is as follows:

### 3.3 Formal definition of a building semantic model

A building contains many different entities, such as equipment in various locations, and these entities are interconnected. A structure that captures this information is called a "building semantic model" and can be interpreted as a KG. The mathematical formalisation of the "building semantic model" is a directed acyclic graph $\mathcal{G} = (V, P, E)$ where:

**Vertices (V)**: Each vertex $v \in V$ represents an entity within the building. This could be a physical location (e.g., a room or a zone served by a single HVAC subsystem), a piece of equipment (e.g. an air temperature sensor or a fan coil unit), or a reference to a time series in the form of a unique key. The actual time series data is typically stored in a separate database.

**Edges (E)**: Each edge $e = (u, p, v) \in E$ represents a predicate $p$ between two vertices $u$ and $v$.

**Predicate (P)**: Each edge $e$ is associated with a predicate $p \in P$ that specifies the type of relationship it represents (e.g., hasPart, has Location, or isPointOf).

### 3.3.1 BTS and LBNL59

**BTS complements LBNL59** due to differences in time and location, as well as the size and complexity of the buildings. While LBNL59 covers a period ending in 2020 in the USA, our dataset spans from 2021 onwards in Australia, offering insights into longitudinal change and different seasonal patterns. Additionally, our dataset includes larger and more complex buildings compared to those in LBNL59.

---

[5]The reason for the discrepancy between the number of timeseries and `Point` is that multiple time series can be associated with the same Point in some instances.

Table 2: **Summary statistics** of the three buildings in our Building TimeSeries (BTS) dataset in comparison with LBNL59 [36, 49]. The table details the count and unique count (in parentheses) for the top-level Brick ontology [7] and the `Point` sub-classes. [5]

| | Count (Unique) | LBNL59 | | BTS_A | | BTS_B | | BTS_C | |
|---|---|---|---|---|---|---|---|---|---|
| **Top Level** | Collection | 0 | (0) | 4 | (2) | 2 | (2) | 8 | (1) |
| | Equipment | 59 | (3) | 547 | (24) | 159 | (25) | 963 | (41) |
| | Location | 73 | (3) | 481 | (9) | 68 | (17) | 381 | (26) |
| | Point | 230 | (11) | 8374 | (126) | 851 | (57) | 10440 | (159) |
| | Timeseries | 337 | | 8349 | | 851 | | 5347 | |
| **Point Subclass** | Alarm | 0 | (0) | 798 | (16) | 5 | (2) | 109 | (8) |
| | Command | 0 | (0) | 363 | (6) | 97 | (5) | 785 | (13) |
| | Parameter | 0 | (0) | 79 | (6) | 36 | (2) | 935 | (17) |
| | Sensor | 144 | (8) | 4396 | (56) | 266 | (25) | 4062 | (68) |
| | Setpoint | 86 | (3) | 772 | (26) | 232 | (16) | 1629 | (41) |
| | Status | 0 | (0) | 1628 | (17) | 110 | (6) | 2187 | (19) |
| | Location | Berkeley, USA | | Undisclosed locations in Australia | | | | | |
| | Start Date | 01-01-2018 | | 01-01-2021 | | 01-01-2021 | | 23-06-2021 | |
| | End Date | 31-12-2020 | | 31-12-2023 | | 31-12-2023 | | 18-01-2024 | |
| | Duration (Days) | 1094 | | 1094 | | 1094 | | 939 | |
| | Size Zipped (GB) | 0.26 | | 8.48 | | 1.31 | | 8.98 | |

**BTS dataset is larger and more diverse.** Each building in BTS includes significantly more timeseries—ranging from double to over twenty times more—resulting in a combined file size approximately 70 times larger when zipped.

The BTS dataset also exhibits greater diversity. Although LBNL59 contains 337 different timeseries, they are composed of only 11 different classes, all classified as either `Sensor` or `Setpoint`. In contrast, the BTS dataset has hundreds of unique `Point` classes including additional categories such as `Alarm`, `Command`, `Parameter`, and `Status`, offering a more comprehensive and varied dataset.

### 3.3.2 Addressing Literature Gaps with BTS Dataset

In Sections 2.2 and 2.3, the importance of scalability and interoperability was underscored, alongside the notable properties exhibited by our datasets, including domain shift, multimodality, imbalance, and long-tailedness. Here, we elaborate on how the BTS dataset effectively addresses these identified gaps in the literature.

**Brick is machine-readable and multimodal.** Consequently, this dataset fuels the research into building-agnostic, interoperable, and scalable software and ML models for building analytics. As a KG, Brick includes text components, facilitating novel research into interactions between KG, LLM and MVTS data.

**Our dataset is from real-world buildings.** This inclusion highlights real-world issues, as illustrated in Figure 1. For instance, the anomalously straight segments in `Air Temp Sensor`, `Outside Air Temp Sensor`, and `Enable Status` during the middle of May might indicate that there are missing values. Additionally, at the end of June, an anomalous data point is observed where the temperature sensors and setpoint limits drop to zero at the same time. It remains unclear if this was intentional, or by accident, or an error. This dataset serves as a test bed to evaluate how ML pipelines can address such issues during inference.

**Domain Shift.** The presence of domain shift complicates transfer learning efforts, as each building exhibits a unique distribution of classes. For instance, in the BTS_A, over half of the timeseries are `sensors`, whereas in BTS_B, this proportion drops to less than a third. Similarly, approximately a third of timeseries in BTS_B are `setpoints`, compared to less than a tenth in BTS_A.

Moreover, individual timeseries within each building demonstrate distinct distributions. As depicted in Figure 1, `Outside Air Temp Sensor` exhibit periodic behavior, leading to a more normal distribution, while `Electrical Power Sensor` display a non-periodic, monotonically increasing

pattern, and `Enable Status` adheres to a Bernoulli distribution. Moreover, as shown in the figures in Appendix B, there is a significant disjoint of ontological classes between buildings; more than half of the classes only appear in one of the buildings only. Therefore, our dataset serves as an ideal dataset for investigating domain shifts.

**Long-Tailed Distributions.** The class distribution in BTS exhibits a long tail as shown in the figures in Appendix B. This means that certain class appear frequently, such as the 1004 instances of `Electrical Power Sensor` across all three buildings (Figure 4), while others are rare, with 10 classes appearing only once in the entire dataset, such as the `Air Differential Pressure Setpoint` location in `BTS_C` (Figure 7). Similarly, the values in some timeseries also follow a long-tailed distribution. For example, `Alarms` are expected to remain at zero most of the time.

# 4 Benchmark

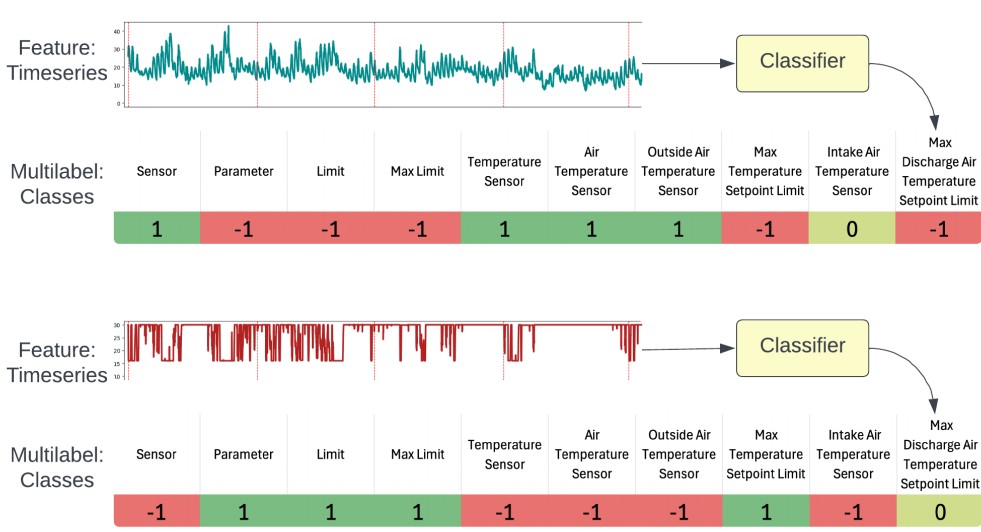

Figure 3: Visualisation of the multi-label timeseries classification task.

To demonstrate the utility of this dataset, we conducted a benchmark on the multi-label timeseries classification task. We picked this task because it highlights the challenges in implementing machine learning model that is interoperable between buildings. We also performed an additional benchmark on a zero-shot forecasting task. The details can be found in Appendix D.

Brick schema [7] was developed to aids in data interoperability across buildings. However, constructing the Brick schema for each building requires expensive and error prone manual expert labor to classifying timeseries data into the correct Brick classes. Past studies [8, 43, 67] have attempted to automate this process with ML relied on private data and did not release their code. This benchmark is the first to address the task using publicly available data. We formulated this task as a multi-label timeseries classification task, where a label will also return true for all super-classes and return as zero for all subclass. More details on this benchmark can be found in Appendix C

## 4.1 Problem Formulation

A datapoint $d = (t, v)$ is an ordered pair where $t \in \mathcal{R}$ is time and and $v \in \mathcal{R}$ is the value. A timeseries $T = \{d_i | 1 \leq i \leq n\}$ is a vector of datapoint of length $n \in \mathcal{Z}^+$. The length of timeseries can varies.

The class `Point` in Brick has $m$ sub-classes, including both direct and indirect sub-classes. In the original dataset, each timeseries is only labeled with a single class. However, we reformulated this as

a multi-label classification task, where a label will also return true for all super-classes and return as zero for all subclass. More formally, $l_j \in \{-1, 0, 1\}$ for $1 \leq j \leq m$ where $l_j = 1$ if timeseries $T$ belongs to the $j^{th}$ subclass of `Point` and also for all of its super-class, $l_j = 0$ for all of its sub-class, and $l_j = -1$ otherwise. For practical purposes, $m$ is not the number of sub-classes of `Point` in the definition, but only those found in our dataset.

The task for each timeseries is to predict if timeseries $T$ belongs in the $j^{th}$ label $l_j = f(T) \, \forall j$. This is visualised in Figure 3.

## 4.2 Results

Table 3: Benchmark results on the multi-label timeseries classification task. Deterministic methods do not have standard deviation.

| Method | Accuracy | | F1 | | mAP | |
|---|---|---|---|---|---|---|
| Zero | 0.8484 | ±N/A | 0.0000 | ±N/A | 0.0000 | ±N/A |
| Mode | 0.8592 | ±N/A | 0.1296 | ±N/A | 0.0990 | ±N/A |
| Random Proportional | 0.8147 | ±0.0001 | 0.1487 | ±0.0002 | 0.1520 | ±0.0001 |
| Random Uniform | 0.4999 | ±0.0002 | 0.1813 | ±0.0002 | 0.1520 | ±0.0001 |
| One | 0.1516 | ±N/A | 0.2234 | ±N/A | 0.1516 | ±N/A |
| LR | 0.2366 | ±N/A | 0.0882 | ±N/A | 0.0497 | ±N/A |
| XGBoost | 0.8593 | ±N/A | 0.2697 | ±N/A | 0.2627 | ±N/A |
| Transformer (default) | 0.7807 | ±0.0139 | 0.3360 | ±0.0116 | 0.3171 | ±0.0078 |
| Transformer (HP tuned) | 0.8052 | ±0.0074 | 0.3615 | ±0.0079 | 0.3489 | ±0.0057 |
| Informer | 0.7627 | ±0.0010 | 0.3162 | ±0.0019 | 0.2849 | ±0.0030 |
| DLinear | 0.7030 | ±0.0042 | 0.2499 | ±0.0020 | 0.2494 | ±0.0010 |
| PatchTST | 0.7534 | ±0.0017 | 0.2981 | ±0.0014 | 0.2721 | ±0.0013 |

Table 3 shows the results. Notice how naive methods achieved very high accuracy but very poor F1 and mean Average Precision (mAP) scores, while deep learning methods obtained slightly better F1 and mAP scores but much poor accuracy. We attribute this to the extreme imbalance in our dataset. All models performed only slightly better than the naive methods, indicating that this is an unsolved problem with significant potential for new discoveries.

Refer to Appendix C for more details about this experiment, including formal problem formulation, more results and other experimental details.

# 5 Limitations

Firstly, the dataset is sourced from only three non-residential buildings in Australia, limiting its geographical diversity. Consequently, models trained on this dataset may not generalize well to residential buildings, or buildings in other regions with different climates, regulations, and building practices. This limitation implies that models should primarily be used for research purposes rather than direct deployment.

Secondly, the anonymization process, essential for privacy, may have removed valuable context-specific information, such as building layouts, occupancy patterns, and operational schedules. This reduction in detail could limit the dataset's applicability for certain analyses. Moreover, despite thorough anonymization efforts, there is no absolute guarantee that personally identifiable information cannot be recovered, particularly when correlated with external datasets.

Finally, as this paper focuses on the dataset rather than benchmarking, the depth of the benchmarks is limited. For example, hyperparameter optimization was not performed.

# 6 Conclusion

In this paper, we introduced the Building TimeSeries (BTS) dataset, addressing the critical gaps in building analytics research by providing a comprehensive, publicly available dataset that spans three buildings over three years, encompassing over ten thousand timeseries data points and 240

unique classes. This dataset is standardized using the Brick schema, ensuring interoperability and consistency across analyses. Additionally, our datasets inherently possess properties of interest to machine learning research, such as domain shift, multimodality, imbalance, and long-tailedness. Our benchmarks on multi-label timeseries classification and zero-shot forecasting tasks demonstrate the dataset's utility in addressing key challenges in building analytics. By making the BTS dataset and our benchmarking code publicly accessible, we aim to facilitate further research in optimizing building performance, ultimately contributing to efforts to mitigate climate change and enhance human well-being.

## Acknowledgments and Disclosure of Funding

This research is supported by the NSW Government through the CSIRO's NSW Digital Infrastructure Energy Flexibility (DIEF) project, funded under the Net Zero Plan Stage 1: 2020-2030.

This project is also funded by the Reliable Affordable Clean Energy for 2030 (RACE for 2030) Cooperative Research Centre.

This research was undertaken with the assistance of resources from the National Computational Infrastructure (NCI Australia), an NCRIS enabled capability supported by the Australian Government.

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

# Appendices

## A   List of Related Works

For convenience, we summarised the works listed in Section 2.1 in Table 4.

Table 4: List of related work.

|  | Datasets |
|---|---|
| Private | HVAC [69, 33, 77, 71, 30, 29, 20], energy use [60, 61], timeseries ontology classification [34, 35, 23, 42, 43, 67], and simulation [76]. |
| Paid | Pecan Street [14]. |
| Upon discretion of the data provider | ecobee [21]. Mortar [22] is intended to be freely available, yet it has limited access due to cloud deployment issues at the time of writing). |
| Static | EUBUCCO [53], PLUTO [17], GBMI [10], Roofpedia [87], HBD3D [9], |
| Corase temporal granularity (more than daily) | CBECS [16], BERTOOL [75], CENED+2 [68], |
| Simulation-based | BEM4CBECS [2, 91, 92, 90], ResStock [84], ComStock [58], CityLearn Challenge Series [81, 54, 57, 56], and hardware-in-the-loop laboratory [65, 64]. |
| Limited scope | SLRHOME [5], LCLD [79], and UCI [78] |
| NILM | Non-intrusive load monitoring (NILM) is task and many dataset have been made for this task check this recent survey [59] that list publicly available dataset. However, since the datasets are only made for this specific task in mind, the scope is limited to only electricity submetering. Other datasets with focus on submetering: BDG [52] and BDG2 [51]. |
| Occupant behaviour | From AshraeOB [18, 47] website: "The ASHRAE Global Occupant Behavior Database aims to advance the knowledge and understanding of realistic occupancy patterns and human-building interactions with building systems. This database includes 34 field-measured occupant behavior datasets for both commercial and residential buildings, contributed by researchers from 15 countries and 39 institutions covering 10 different climate zones. It includes occupancy patterns, occupant behaviors, indoor and outdoor environment measurements." |
| **Comprehensive** | Lawrence Berkeley National Laboratory building 59 (LBNL59) [36, 49] and **BTS (ours)** https://github.com/cruiseresearchgroup/DIEF_BTS. |
| Other lists | A review paper on NILM [59], a review paper on buildings at urban scale [72], a review paper on energy flexibility datasets [44], a review paper on building and energy dataset [40], and the Building Data Genome Directory [39]. |

## B   Visualisation of Domain Shift and Long-tail Distribution in Our Datasets.

We visualise the domain shift by comparing the different distributions of classes between buildings. We visualise the that the distributions of classes have long-tails by plotting the histogram. These are shown in Figure 4, 5, 6, and 7. The relevant discussions can be found in Section 2.3 and 3.3.2.

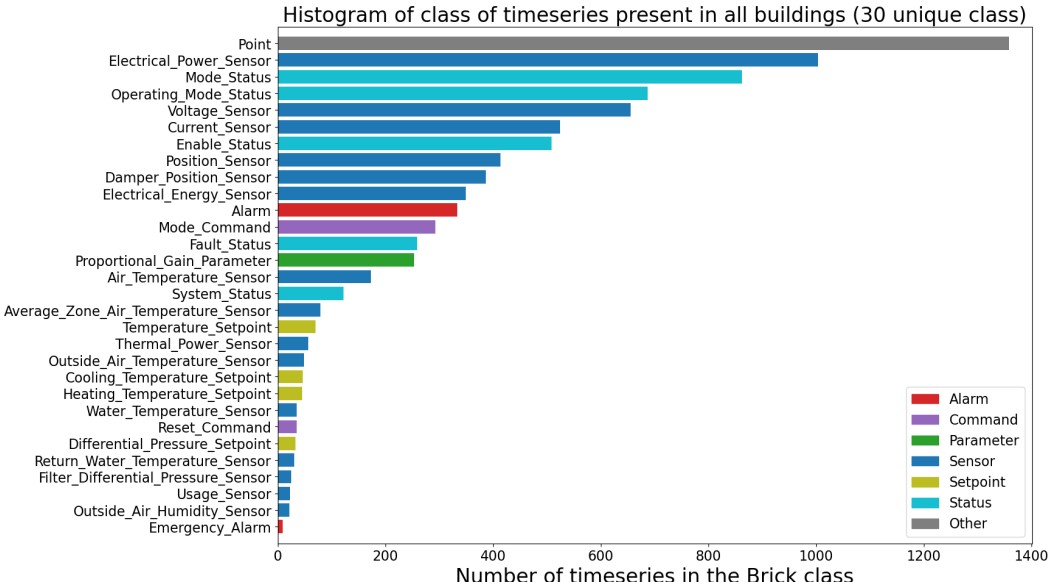

Figure 4: Histogram of class of timeseries by buildings.

## C    Multi-label Timeseries Classification: More details

### C.1    Data Pre-processing

Each timeseries are cut into shorter chunk of either 2/4/8 weeks. The reason is to enable analysis of accuracy against various length of the timeseries. Those with too few datapoint, less than 1 per day, are removed. Due to great ranges of values, they are scaled using symmetric log first, and then standard scaling. The symmetric log function is defined as follows:

$$v' = \begin{cases} 9 + \log_{10}(v) & \text{if } v > 10 \\ -9 - \log_{10}(-v) & \text{if } v < -10 \\ v & \text{otherwise} \end{cases}$$

### C.2    Development and Test Partition

The partition is done by time and buildings. The reason for this partition strategy is to evaluate the performance in the future, and in different buildings. The development partition consist of the first four months of BTC_A and the first year of BTC_B. The development partition is randomly split into training and validation with a 80% and 20% ratio respectively. The remaining data are set to the testing partition.

### C.3    Feature Extraction

Depending on whether the models are made generic classification (LR, RF, and XGBoost) or deep learning models specialised for timeseries, a different feature extraction method were used. For generic models, we extract the following global features: mean, standard deviation, skew, kurtosis, root mean square, minimum, maximum, the three quartiles, and average duration between data points. For timeseries algorithm, we aggregate the timeseries into four hour slots and extract the maximum, mean, standard deviation, and number of datapoints within each slot.

### C.4    Model Training

We used binary cross-entropy (BCE) loss, treating every single label as binary, and applied additional extra weight to the positive samples proportionally. The maximum number of epochs was set to 100,

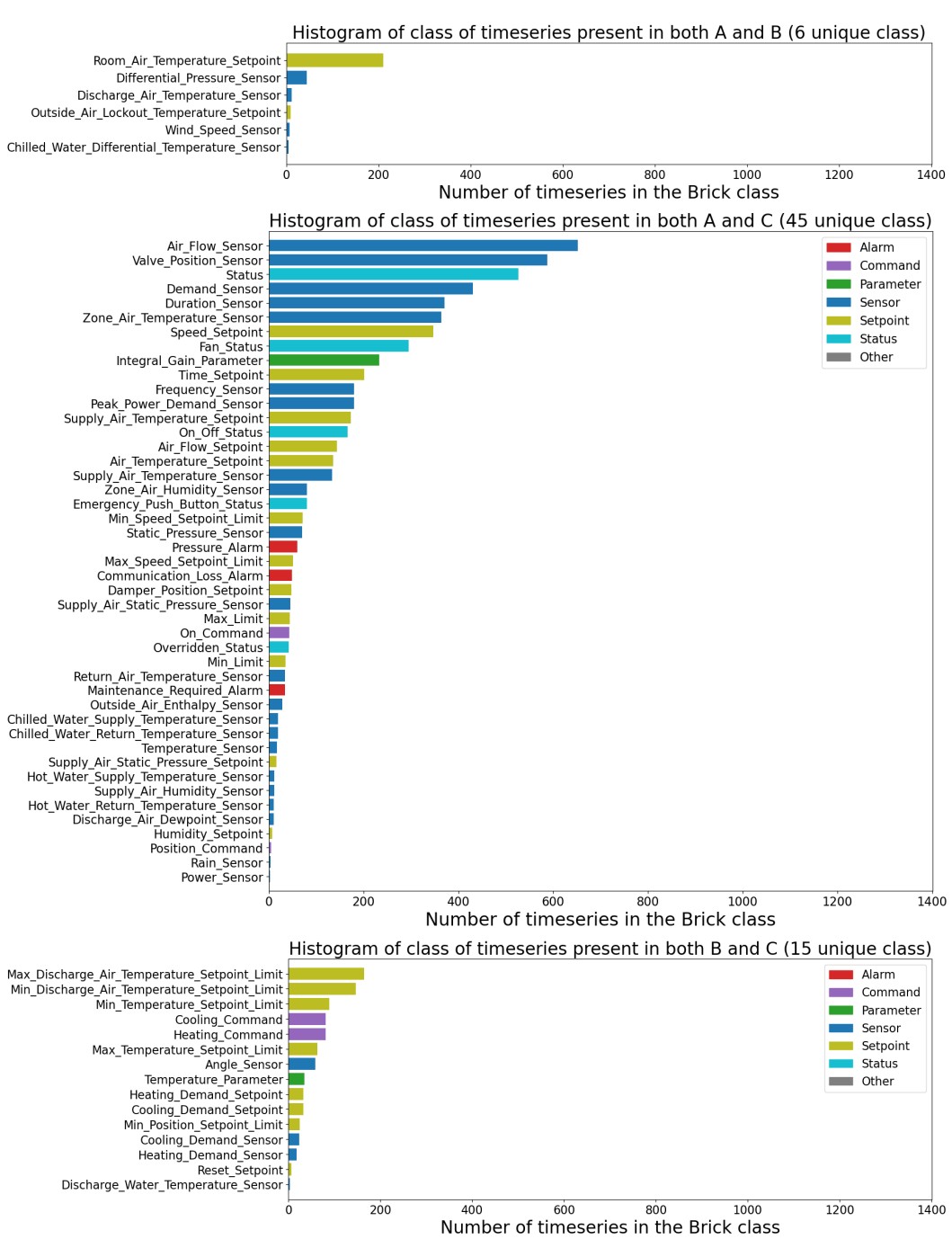

Figure 5: Histogram of class of timeseries by buildings, continued.

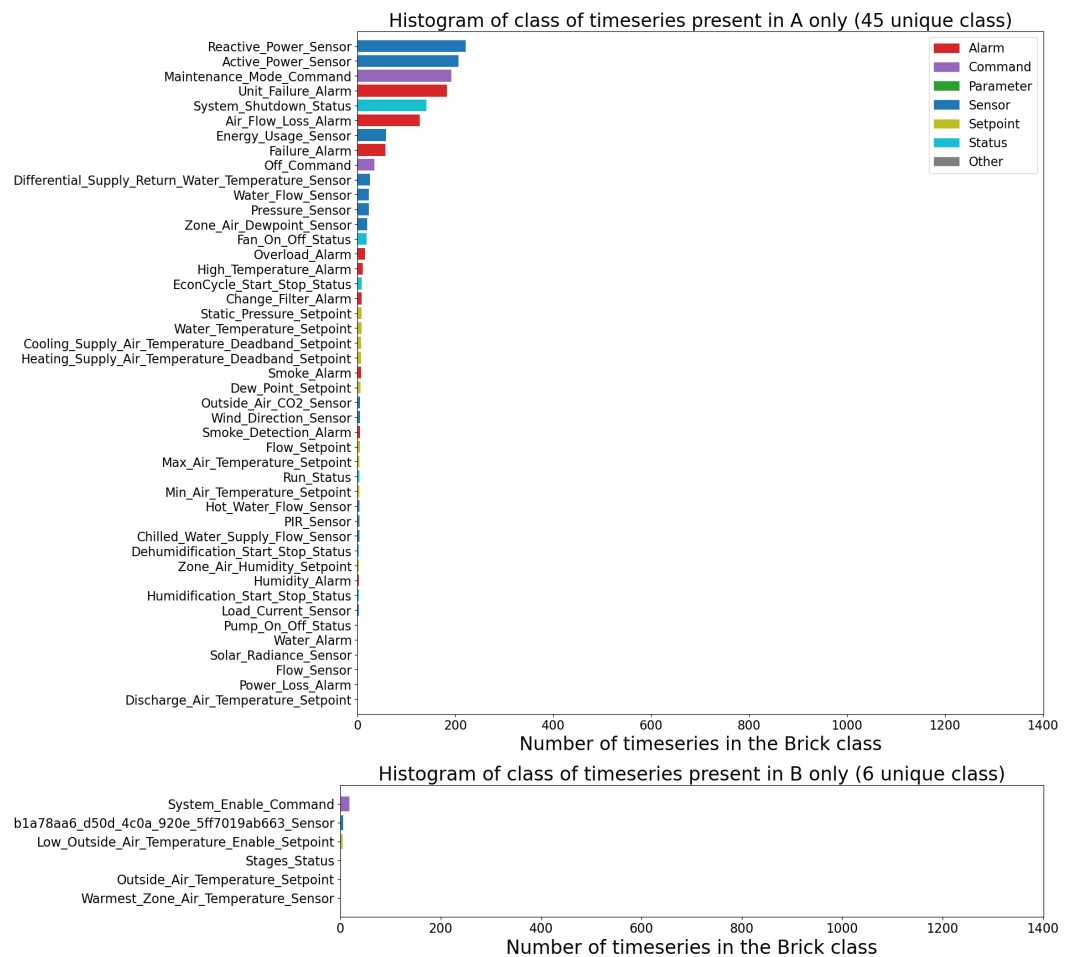

Figure 6: Histogram of class of timeseries by buildings, continued.

with a patience of 30 epochs for early stopping. The learning rate was set to 0.01, and we used the ReduceLROnPlateau strategy with a patience of 10 epochs. The optimizer was Rectified Adam (RAdam). For deep learning methods, we adapted the TSLib code [88] from their official GitHub repository `https://github.com/thuml/Time-Series-Library`. The batch size for each method was adjusted to fit memory. Our implementations, including our hardware setup, are available on the GitHub repository for this project `https://github.com/cruiseresearchgroup/DIEF_BTS`.

### C.5 Baselines

We use four naive baselines that does not take the features into account:

- **Zero.** The model output negative prediction on all labels.

- **Random Uniform.** The model based the prediction on a coin flip (50/50)

- **Random Proportional.** The model based the prediction randomly, but according to the proportion each label appears on the training data.

- **Mode.** The most common class was `Sensor`. So the model predict `Sensor` all the time.

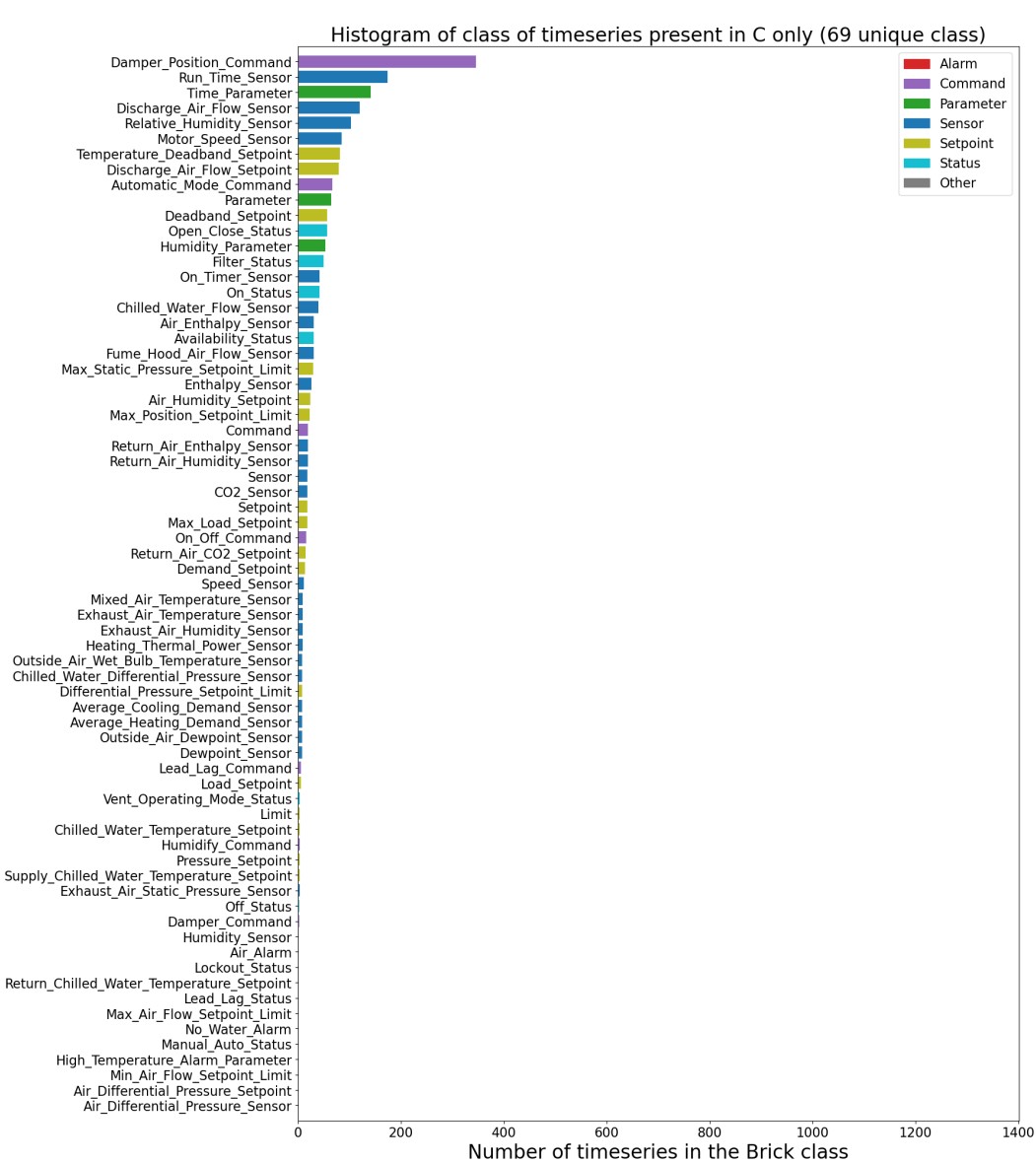

Figure 7: Histogram of class of timeseries by buildings, continued.

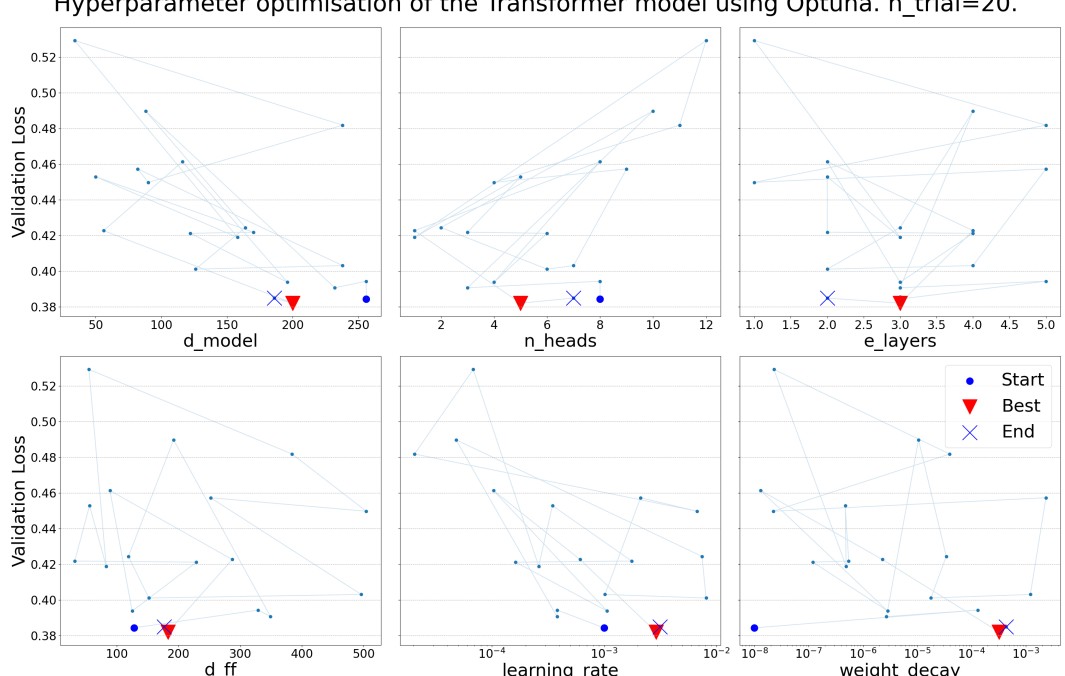

Figure 8: Visualisation of the hyperparameter tuning of the Transformer model.

## C.6 Hyperparameter Tuning

# D Zero-shot Forecasting Across Buildings

The advent of building digitalization presents significant opportunities for leveraging deep learning methods in building management systems for accurate forecasting. In practical applications, it is crucial for well-trained models to be applicable across diverse building scenarios without retraining costs. However, specific building constraints, operational variances, functionality differences, and data heterogeneity pose significant challenges in real-world settings. As shown in Table 2, models must adapt to dynamic ontology changes when applied to different buildings. Previous studies often rely on identical features and well-processed data, not reflecting the complexity of real-world scenarios. LBNL59, involving only one building, is insufficient for transfer learning studies. This study establishes a baseline for zero-shot forecasting using the BTS multivariate time series.

## D.1 Problem Formulation

Suppose we have dataset $D \in \mathcal{R}^{N \times K}$ with $N$ IoT points and $K$ timesteps. Each data point is denoted as $d_{N,k} = D_{N,k:k+S}$, where $S$ is the sequence length of the historical data. Detnote the forecasting model as $h(\cdot)$. The multi-step forecasting problem is formalized as follows: $h(d_{N,k}) = d_{N,k+S+H}$, where $H$ is the forecasting horizon. In zero-shot forecasting, the model is trained and tested across different datasets. In this study, $S = 12, H = 12$.

## D.2 Data Pre-processing

This study utilizes a 1-month training dataset spanning from 00:00:00 on 01/07/2022 to 00:00:00 on 01/08/2022, with irregular data resampled to a 10-minute granularity and then standardization. The historical window and forecast horizon are set to 12 time steps, equivalent to 2 hours. A model trained on one dataset is evaluated across all buildings for the same period. For each dataset, a subset of IoT points is selected for training based on the criterion $N_{\text{unique}}/N_{\text{sample}} > \eta$ where $\eta = 0.1$. This feature selection results in 133, 710, and 2025 IoT points for the three respective datasets.

## D.3   Baselines

DLinear[93], PatchTST[55], Informer[96] and iTransformer[48] as backbone models are employed for this benchmark study.

## D.4   Model Training

While employing DLinear for training, we treat this task as a multivariate forecasting task. Models are fed by all the IoT points data and expect to forecast the corresponding values of these IoT points. Considering that certain Transformer-based backbone models that involve a conventional embedding layer, such as iTransformer, Informer, and PatchTST, do not support changes in input channels between training and testing sets, we handle the task as an univariate forecasting problem, treating each IoT point equivalently. Similar to the multi-label classification task, the code is modified based on TSLib[88] Github repository. The training process employs the Adam optimizer with a learning rate of 0.01 and Mean Square Error (MSE) loss, and a learning rate scheduler is applied. Training is capped at 20 epochs with an early stopping patience of 3 epochs. All experiments are conducted on the NCI Gadi server utilizing 4 V100 GPUs.

## D.5   Metrics

Baseline performance is evaluated using Mean Absolute Error (MAE) and Symmetric Mean Absolute Percentage Error (SMPAE) averaged by IoT points. Following the above-mentioned notation, the mathematical definitions are as follows:

$$MAE = \frac{1}{N} \sum_{n=1}^{N} |\hat{y_n} - y_n|$$

$$SMAPE = \frac{100\%}{N} \sum_{n=1}^{N} \frac{|\hat{y_n} - y_n|}{|\hat{y_n}| + |y_n|}$$

$$R^2 = 1 - \sum_{n=1}^{N} (y_n - \hat{y}_n)^2 / \sum_{n=1}^{N} (y_n - \bar{y})^2$$

where $\hat{y}_n, y_n, \bar{y}_n$ are the multi-step prediction, ground truth, and mean for the evaluated model.

## D.6   Main Results

Table 5 presents the Symmetric Mean Absolute Percentage Error (SMAPE) and $R^2$ scores for four baseline models in this task, with diagonal values omitted. PatchTST and DLinear consistently outperform the other models, balancing higher $R^2$ scores with lower SMAPE values. However, the overall performance highlights the complexity and challenges inherent in zero-shot forecasting, indicating significant scope for further research and improvement.

## D.7   Detailed Results with Standard Deviations

Table6-8 shows the mean and standard deviation values about MAE, SMAPE, and $R^2$ for the multi-step zero-shot forecasting.

Table 5: Benchmark results on the zero-shot forecasting task. The columns refer to the training set, whereas the row represents the testing set.

| | | BTS-A | | BTS-B | | BTS-C | |
|---|---|---|---|---|---|---|---|
| | | MAE | SMAPE | MAE | SMAPE | MAE | SMAPE |
| Previous Day Persistence | | 0.5377 | 48.1539 | 0.4976 | 43.2985 | 0.5458 | 45.7014 |
| Previous Week Persistence | | 0.6190 | 57.2713 | 0.5918 | 51.3867 | 0.6499 | 58.1922 |
| BTS-A | DLinear | N/A | | 0.4324 | 35.9846 | 0.4262 | 36.2734 |
| | PatchTST | N/A | | 0.3748 | 29.2570 | 0.3712 | 29.5552 |
| | Informer | N/A | | 0.5968 | 49.2217 | 0.5920 | 51.9745 |
| | iTransformer | N/A | | 0.4026 | 31.1924 | 0.3842 | 30.1102 |
| BTS-B | DLinear | 0.4940 | 41.2264 | N/A | | 0.4206 | 35.3121 |
| | PatchTST | 0.4575 | 36.7689 | N/A | | 0.3711 | 29.2135 |
| | Informer | 0.5233 | 45.9279 | N/A | | 0.4592 | 39.7068 |
| | iTransformer | 0.4783 | 37.5907 | N/A | | 0.3901 | 29.9940 |
| BTS-C | DLinear | 0.4858 | 40.7421 | 0.4158 | 34.1473 | N/A | |
| | PatchTST | 0.4542 | 36.9451 | 0.3723 | 28.9325 | N/A | |
| | Informer | 0.5213 | 46.6112 | 0.4602 | 39.7162 | N/A | |
| | iTransformer | 0.4859 | 39.5158 | 0.4262 | 32.6550 | N/A | |

Table 6: Mean Absolute Error (MAE) on the zero-shot forecasting task. The columns refer to the training set, whereas the row represents the testing set.

| | Method | BTS_A | BTS_B | BTS_C |
|---|---|---|---|---|
| BTS_A | DLinear[93] | N/A | 0.43243±0.16060 | 0.42617±0.19525 |
| | PatchTST [55] | N/A | 0.37480±0.06301 | 0.37480±0.06301 |
| | Informer [96] | N/A | 0.59679±0.04698 | 0.59196±0.05424 |
| | iTransformer [48] | N/A | 0.40257±0.06487 | 0.38416±0.07446 |
| BTS_B | DLinear[93] | 0.49398±0.21579 | N/A | 0.42059±0.20122 |
| | PatchTST [55] | 0.45745±0.08428 | N/A | 0.37106±0.07449 |
| | Informer [96] | 0.52329±0.06606 | N/A | 0.45922±0.05966 |
| | iTransformer [48] | 0.47830±0.08542 | N/A | 0.39099±0.07722 |
| BTS_C | DLinear[93] | 0.48582±0.22002 | 0.41582±0.17401 | N/A |
| | PatchTST [55] | 0.45413±0.08338 | 0.37227±0.06339 | N/A |
| | Informer [96] | 0.52133±0.06237 | 0.46022±0.05043 | N/A |
| | iTransformer [48] | 0.48588±0.08002 | 0.42620±0.06586 | N/A |

Table 7: Symmetric Mean Absolute Percentage Error (SMAPE) on the zero-shot forecasting task. The columns refer to the training set, whereas the row represents the testing set.

| | Method | BTS_A | BTS_B | BTS_C |
|---|---|---|---|---|
| BTS_A | DLinear[93] | N/A | 35.98461±15.47196 | 36.27335±18.34376 |
| | PatchTST [55] | N/A | 29.25704±5.03140 | 29.55517±6.07105 |
| | Informer [96] | N/A | 49.22169±2.54525 | 51.97452±4.25621 |
| | iTransformer [48] | N/A | 31.19242±5.23906 | 30.11023±5.97160 |
| BTS_B | DLinear[93] | 41.22638±18.84817 | N/A | 35.31209±18.23204 |
| | PatchTST [55] | 36.76894±6.63363 | N/A | 29.21348±5.96805 |
| | Informer [96] | 45.92792±6.15185 | N/A | 39.70681±5.37708 |
| | iTransformer [48] | 37.59074±6.54195 | N/A | 29.99402±6.02286 |
| BTS_C | DLinear[93] | 40.74205±19.53859 | 34.14733±16.12281 | N/A |
| | PatchTST [55] | 36.94508±6.74060 | 28.93252±5.03300 | N/A |
| | Informer [96] | 46.61115±6.07310 | 39.71622±4.55301 | N/A |
| | iTransformer [48] | 39.51578±6.64577 | 32.65497±5.24526 | N/A |

Table 8: $R^2$ score on the zero-shot forecasting task. The columns refer to the training set, whereas the row represents the testing set.

| | Method | BTS_A | BTS_B | BTS_C |
|---|---|---|---|---|
| **BTS_A** | DLinear[93] | N/A | 0.54196±0.12989 | 0.53206±0.09756 |
| | PatchTST [55] | N/A | 0.51219±0.16793 | 0.51258±0.05317 |
| | Informer [96] | N/A | 0.32122±0.18004 | 0.32153±0.05191 |
| | iTransformer [48] | N/A | 0.46723±0.17016 | 0.48543±0.05315 |
| **BTS_B** | DLinear[93] | 0.43686±0.09253 | N/A | 0.52964±0.09715 |
| | PatchTST [55] | 0.40926±0.03239 | N/A | 0.50624±0.05375 |
| | Informer [96] | 0.39893±0.02753 | N/A | 0.47109±0.04673 |
| | iTransformer [48] | 0.36844±0.03443 | N/A | 0.46792±0.05684 |
| **BTS_C** | DLinear[93] | 0.44519±0.09250 | 0.54543±0.12879 | N/A |
| | PatchTST [55] | 0.41773±0.03099 | 0.51411±0.17089 | N/A |
| | Informer [96] | 0.41886±0.02556 | 0.48993±0.13881 | N/A |
| | iTransformer [48] | 0.37250±0.03034 | 0.42437±0.17611 | N/A |

Models trained on BTS_A exhibit poorer cross-building forecasting results. This can be attributed to the greater complexity of BTS_A compared to BTS_B and BTS_C. BTS_A includes more heterogeneous series and entity types (BTS_A has 42 entities, where BTS_B and BTS_C have 16 and 31 entities respectively in the task training data), which introduces additional noise that impacts accuracy.

The evaluation metrics, MAE and SMAPE, indicate that PatchTST outperforms other baselines, while $R^2$ scores suggest that DLinear is superior. However, DLinear also shows a higher standard deviation. This indicates that DLinear effectively captures linearity in sequential data, leading to accurate predictions for IoT points with strong linear relationships. Conversely, it struggles with complex inherent dependencies, resulting in poorer performance on datasets with such characteristics.

The overall scores indicate significant potential for improvement. Considering the comprehensive metadata scope provided by the BTS dataset, future work can leverage knowledge graphs to enhance data modality. This approach could improve the accuracy and robustness of deep learning models in zero-shot forecasting.

