# Supplementary Materials:
# BTS: Building Timeseries Dataset:
# Empowering Large-Scale Building Analytics

# 1 Appendices

## 2 E  Extra information for NeurIPS submission

Note that we are planning to use this dataset for a competition, so we are only releasing a small snippet at this moment. Information about the competition is not available yet, because it is still in the planning stage. We will release the full dataset after the competition is completed.

- Dataset documentation and intended uses. Data card is available on the GitHub: `https://github.com/cruiseresearchgroup/DIEF_BTS/blob/main/BTS_DataCards.md`

- URL to website/platform where the dataset and benchmark can be viewed and downloaded: `https://github.com/cruiseresearchgroup/DIEF_BTS`

- URL to Croissant metadata will not be made available, as typical with spatiotemporal time-series dataset. See previous accepted dataset: `https://github.com/liuxu77/LargeST`.

- The author hereby state that we bear all responsibility in case of violation of rights.

- The data license is CC BY 4.0 and the code license is the MIT License.

- The data is hosted on FigShare `https://figshare.com/articles/dataset/BTS_Building_Timeseries_Dataset_Empowering_Large-Scale_Building_Analytics_TRAIN_ONLY_/25912180` while the code are hosted on GitHub `https://github.com/cruiseresearchgroup/DIEF_BTS`, both are reliable, popular, and reputable platforms for hosting datasets and code.

- The download link to the data and the metadata is posted on `https://github.com/cruiseresearchgroup/DIEF_BTS` once it is made available.

- Data format: Some file are compressed using ZIP. The timeseries dataset are saved as pickle `.pkl` files (`https://docs.python.org/3/library/pickle.html`). The building metadatas follow the Brick schema (`https://brickschema.org/`), and saved as turtle `.ttl` files (`https://www.w3.org/TR/turtle/`).

- Long-term preservation is ensured by uploading it to the `https://figshare.com/` data repository.

- Structured metadata is automatically generated by and available at the FigShare page: `https://figshare.com/articles/dataset/BTS_Building_Timeseries_Dataset_Empowering_Large-Scale_Building_Analytics_TRAIN_ONLY_/25912180`.

- DOI to the dataset :`10.6084/m9.figshare.25912180`.