# OpenReview forum: "Building Timeseries Dataset: Empowering Large-Scale Building Analytics"
_NeurIPS.cc/2024/Datasets_and_Benchmarks_Track — NeurIPS 2024 Track Datasets and Benchmarks Poster_

### Official Review · Reviewer_oLK5 · 2024-07-03
**Summary of Building Timeseries Dataset**

**Rating:** 8
**Confidence:** 5
**Correctness:** It is a dataset and it is constructed…

**Review:**

The paper is of high quality, written with clarity and proposes a new building time series (BTS) dataset which is an advanced version of the other available datasets addressing building analytics research. The research in building analytics is crucial because the buildings account for a third of global energy usage and a quarter of global carbon emissions. Optimizing building performance has the potential to mitigate climate change.

Its significance is that standardized time series data aids in scalability and deployability to build generalized multi-zone environments. The dataset enables research as it contains the Brick schema which is a knowledge graph on building metadata, describing relationships between the time series in the multivariate time series.  The BTS dataset being unbalanced and exciting distributions long tails poses analyses challenges which would lead to the development of innovative techniques.

Authors have taken care of the privacy concerns by anonymizing the attributes in the dataset which is an important consideration. The benchmark results on two machine learning models well demonstrate the usability of the dataset.

**Strengths:**

There are only a handful of datasets which allow for the building analytics.  The new BTS dataset complements the building analytics research towards building-agnostic, interoperable, and scalable software and ML models for building analytics. The dataset is from real-world buildings representing the features, anomalies from real time observations of different sensors and other environments. The dataset may also serve as a testbed to evaluate how ML pipelines can address such issues during inference.   The Paper also explores the benchmarking performance in two ways: time series ontology classification and zero-shot forecasting to understand the challenges in implementing machine learning model interoperability between buildings.

**Additional Feedback:**

In Table 2 caption, it is mentioned “ There are several reasons why the number of time series and Point does not match.” I think this could be explained with reasons including why the number of time series and points does not match.

**Clarity:**

The paper is well written. The dataset and its features are clearly explained along with its comparison with other datasets and variables

**Documentation:**

The dataset includes detail on the data collection process and organization, availability and maintenance and ethical and responsible use. The paper includes the URL for access to the dataset and a hosting, licensing and maintenance plan.

**Ethics:**

I do not see significant ethical concerns. Authors have anonymized the dataset and removed the concerned variables from the dataset as well.

**Limitations:**

Authors have mentioned the limitations of their dataset in a detailed way. I liked that they included the usage suggestions that the dataset should be used for research purposes not for deployment purposes. It is because the dataset is limited by the geographical boundary and does not represent different climates,  regulations, etc. Due to the anonymization of the datasets, some valuable context specific information has been removed. This would limit the applicability of the dataset analysis.
I do not see any potential negative societal impact.

**Opportunities For Improvement:**

In Table 2 caption, it is mentioned “ There are several reasons why the number of time series and Point does not match.” I think this could be explained with reasons including why the number of time series and points does not match.

**Relation To Prior Work:**

The work clearly differentiates itself from previous contributions, the improvements over previous datasets and the limitations over them.

**Summary And Contributions:**

Building analytics involves the automated adjustment of building operations to minimize emissions and costs, optimize energy usage, and enhance indoor environmental quality and occupant experience, including comfort, health and safety. It is crucial because the buildings account for a third of global energy usage and a quarter of global carbon emissions. Optimizing building performance has the potential to mitigate climate change.

This paper introduces real world anonymized data on building operations from three buildings in undisclosed Australian locations called Building time series (BTS) dataset which is a diverse data aimed at addressing the gaps in building analytics research. The dataset includes time series data and building metadata standardized according to the Brick Schema. This serves as a good alternative to the only comparable dataset, LBNL59, limited to single buildings as BTS fills the void by providing data from three diverse buildings, spanning a three-year period and encompassing over ten thousand time series data points and 240 unique ontologies.

BTS dataset possess properties relevant to machine learning research, including domain shift, multimodality, imbalance and long-tailedness. To demonstrate the utility of this dataset, the authors have conducted benchmarks on two machine learning model interoperability tasks. BTS dataset advances  the pursuit of optimizing building performance, ultimately aiding efforts to mitigate climate change  and enhance human flourishing.

---

> ### Author Rebuttal · Authors · 2024-08-17
>
> > In Table 2 caption, it is mentioned “There are several reasons why the number of time series and Point does not match.” I think this could be explained with reasons including why the number of time series and points does not match.
>
> We appreciate the reviewer's suggestion to provide further clarification regarding the mismatch. The reason for this discrepancy is that multiple time series can be associated with the same Point in some instances. We will revise the caption to explicitly include this explanation. We thank the reviewer for helping us to improve the clarity of our manuscript.

---

> ### Author Response · Authors · 2024-08-21
>
> Dear Reviewer, we sincerely appreciate your encouraging review of our paper. We have addressed your clarifying question in the revised manuscript. Given the minor nature of the change, we are hopeful that our response meets with your continued approval. We also value the opportunity to engage in further discussion with you and hope to utilize the remaining time for a productive dialogue. Also note that, based on suggestions from other reviewers, we have incorporated __**additional experiments**__ into our work. For more details, we kindly invite you to check the general rebuttal or the specific thread dedicated to reviewer 1N9M. Thank you again for your kind attention and valuable feedback.

---

> > ### Comment · Reviewer_oLK5 · 2024-08-22
> >
> > Thank you for addressing my comments. I have no additional feedback/comments.

---

### Official Review · Reviewer_UAax · 2024-07-12
**An interesting time series  dataset that will benefit the academia and can help combating global warming**

**Rating:** 7
**Confidence:** 3
**Correctness:** The metrics and methodologies are sta…
**Clarity:** The paper is well-written and easily …

**Review:**

BTS provides data from three diverse buildings, spanning a three-year period and encompassing over ten thousand timeseries data points and 240 unique ontologies.  This kind of long term free public datasets are quite rare and it would benefit the academia greatly in studying the methods for long term time series methods, and it is an ideal dataset to study domain shift / adaptation and multimodal learning with knowledge graphs.   BTS is real-world instead of simulation, and the dataset is from three buildings of diverse ontologies, allowing the exploration various transfer learning techniques to ensure that machine learning models are interoperable between buildings.
Two benchmarks, i.e., timeseries ontology classification and zero-shot  forecasting, were conducted in this work.
This dataset helps designing zero-energy buildings strategy, which is probably at the rightest moment when the world is getting only warmer by year.
On top of existing data center electricity demand data, green building data would be a big plus to bring attention to reduce the overall emission.
The paper is well written with good readability.

**Strengths:**

BTS is real-world instead of simulation, and the dataset is from three buildings of diverse ontologies, allowing the exploration various transfer learning techniques to ensure that machine learning models are interoperable between buildings.
BTS contains ten thousand time series data points, featuring a diverse array of 240 unique ontologies,  surpassing the ontological breadth of LBNL59 by more than threefold.
BTS enables researchers to explore buildings with more intricate analytics  setups, facilitating deeper insights into building dynamics and performance. Furthermore, the meta data are standardized using the popular Brick schema, ensuring consistency and compatibility across analyses.

**Additional Feedback:**

none

**Documentation:**

clean and clear documentation

**Ethics:**

only positive impact to the academic and industrial communities can be thought of

**Limitations:**

The dataset is from a very unique niche of applications.  It could bring bias if being used for other geolocation or different type of buildings.

**Opportunities For Improvement:**

The anomalous data points observed in the datasets are not looked into or investigated，it will be very hard for researchers outside of Australia to investigate whether it is intentional, or by accident, or an error, which could be a big hurdle to eventually facilitate the real-world application of scheduling system for a green building, without any knowledge of the sources of the anomalies.  It is advised for the authors to find out more information from the people managing the buildings.

**Relation To Prior Work:**

BTS complements LBNL59 due to differences in time and location, as well as the size and complexity of the building and BTS dataset is larger and more diverse.
LBNL59 has 11 different ontologies and BTS has hundreds of unique Point ontology including additional categories.
There is a progression from publicly available datasets with narrow focuses (e.g., energy load and temperature) over years to the proposed Building TimeSeries (BTS) dataset, which is noted for its comprehensive scope, capturing detailed and diverse aspects of building operations comprehensively with 215 unique classes.

**Summary And Contributions:**

The paper introduces Building TimeSeries (BTS) dataset, which includes diverse time-series data from buildings over a three-year period, aimed at enhancing research in building analytics. The dataset promotes interoperability in building analytics by adhering to the Bricks schema and supports tasks like time-series ontology classification and zero-shot forecasting.   BTS is a comprehensive and publicly available resource, encompassing over ten thousand timeseries data points and 240 unique ontologies. It can contribute to climate change mitigation and improved human well-being.

---

> ### Author Rebuttal · Authors · 2024-08-17
>
> > The anomalous data points observed in the datasets are not looked into or investigated, it will be very hard for researchers outside of Australia to investigate whether it is intentional, or by accident, or an error, which could be a big hurdle to eventually facilitate the real-world application of scheduling system for a green building, without any knowledge of the sources of the anomalies. It is advised for the authors to find out more information from the people managing the buildings.
>
> We appreciate the reviewer's concern regarding the anomalous data points observed in the datasets. Anomalies are an inherent part of real-world building operations and that machine-accessible labels or explanations for such anomalies are often unavailable in practice.  This dictate the need for robust and resilient solutions to be written to endeavour to handle these anomalies.
>
> In the same way that the topology of a building or types of points may differ from building to building, the anomalous or missing data will also differ. In the spirit of redeployability and ability to transfer learning between buildings, a practical solution to this could utilise the ontological point descriptions to inform the reasonable values expected for the type of point and identifier anomalies in this fashion. Our decision to publish the data "as is" aligns with the need for robust and resilient algorithms that can handle these anomalies without relying on additional information.

---

> ### Author Response · Authors · 2024-08-21
>
> Dear Reviewer, we sincerely appreciate your encouraging review of our paper. We have addressed your clarifying question in the revised manuscript. Given the minor nature of the change, we are hopeful that our response meets with your continued approval. We also value the opportunity to engage in further discussion with you and hope to utilize the remaining time for a productive dialogue. Also note that, based on suggestions from other reviewers, we have incorporated __**additional experiments**__ into our work. For more details, we kindly invite you to check the general rebuttal or the specific thread dedicated to reviewer 1N9M. Thank you again for your kind attention and valuable feedback.

---

> > ### Comment · Reviewer_UAax · 2024-08-22
> >
> > Thanks for addressing my comments and I have no additional feedback/comments.

---

### Official Review · Reviewer_eVsz · 2024-07-24

**Rating:** 6
**Confidence:** 4
**Correctness:** I believe the dataset is constructed …

**Review:**

Strengths
---

* Publicly accessible and comprehensive (in terms of lots of sensors measuring all kinds of things) real-building performance data is hard to come by. This dataset should be practically useful for practitioners that want to apply ML for buildings analytics challenges.
* Multimodal timeseries and graph datasets are scarce in the ML community. This dataset will help elevate research on these two modalities.
* The code and data are already available, and a competition is being planned, which inspires confidence that the dataset will be maintained.

Weaknesses and opportunities for improvement
---

* While an increase from 1 to 3 buildings is an improvement over the previous related dataset, I think it is still too few buildings to  be impactful for transfer learning research and zero-shot generalization research.
* The paper quality could be significantly improved. A lot of the key details are loosely described in the paper and/or relegated to the appendix instead of being concisely presented in the main text (which would help improve clarity too). For example, the dataset preprocessing steps, featurization, dataset class statistics, and mathematical description of the data and use cases are all critical pieces of a NeurIPS Datasets submission. The paper needs proofreading---there are many typos and vague statements.
* While I really like the multimodal timeseries and knowledge graph aspect of the dataset, I felt that the KG aspect of the dataset was not introduced clearly. I don't think "ontology" is used properly throughout the paper (e.g., instead of 240 "ontologies", did the authors mean "entities"?). A formal, mathematical description of the KG and how it is aligned with the timeseries would help inform future multimodal research endeavors.
* The benchmarking part of the paper is not very thorough. I acknowledge that properly conducting benchmark experiments on a new dataset can be very resource (time, compute, etc.) intensive. I would suggest presenting one task comprehensively (with a set of carefully selected and hyperparameter-tuned baselines). For example, the F1 scores in Table 3 are hardly better than random. Would a simple grid search over hyperparameters have improved the best model significantly (the Transformer)?

The Github repository indicates a competition for the two benchmark tasks are planned and so various splits of the data are being withheld. In general, I am supportive of this practice, since it helps prevent overfitting to the test set. However, this seems a bit odd in this case, since the main contribution of this paper is the dataset, not the benchmark tasks.

**Strengths:**

The strengths of the work are detailed above. To summarize, the dataset should be useful for those interested in applying ML to buildings analytics tasks. The dataset has unique characteristics, such as multimodal timeseries and graph data. Code and data are being shared with the public to aid reproducibility and motivate advances.

**Additional Feedback:**

Editing suggestions

* Define ontology
* Table 1: Typo ("BGD2"). Define "Unique Class"
* L166: "...sensors, power, water, and gas meters, and *other devices*..."---what other devices?
* L169: "...for points within the model"---define points
* L173: "...to allow evaluation of various different cleaning algorithm"---which algorithms?
* L190: "...our dataset includes larger and more complex buildings"---can this be quantified?
* Table 2: What is meant by "Count" and "(Unique)"? "There are several reasons why the number of timeseries and Point does not match"---what are the reasons?
* L205: "LLM" is not defined
* L236: "...was developed to *aid* in..."
* L241-242: define super-class and subclass

**Clarity:**

Repeating my previous comment here:
The paper quality could be significantly improved. A lot of the key details are loosely described in the paper and/or relegated to the appendix instead of being concisely presented in the main text (which would help improve clarity too). For example, the dataset preprocessing steps, featurization, dataset class statistics, and mathematical description of the data and use cases are all critical pieces of a NeurIPS Datasets submission. The paper needs proofreading---there are many typos and vague statements.

**Documentation:**

I believe there is enough detail for the dataset to be useful, although there is room for improvement. Dataset documentation (such as a Datasheet) is lacking, and a better discussion of intended uses would be helpful. Some details such as how the data is organized into various pkl files and ttl files, and how to interpret these files, are missing from the paper. The data itself (at least, what is released for the competition so far) is accessible via a URL. The dataset has a DOI and license (CC BY 4.0). A maintenance plan is not discussed in any detail.

**Ethics:**

No. The energy timeseries data has been anonymized.

**Limitations:**

The authors discussed various limitations of the work in Section 5. They acknowledge the lack of depth of the benchmarking section. However, the use of the dataset for a competition about the two benchmark tasks suggests that the benchmark section should be more rigorous.

**Opportunities For Improvement:**

Opportunities to improve the work include:

* Proofreading and revising. This includes making the presentation more concise and clear by improving the description of key dataset details currently relegated to the appendix.
* Discussing more potential uses-cases for the dataset would be helpful.
* To make the benchmark section more rigorous, more baselines can be included (in addition to hyperparameter tuning, which the authors acknowledge in Section 5). For example, the rule-based persistence baselines in [1] are a common and powerful baseline to include for short-term zero-shot forecasting on buildings.

[1] Emami, Patrick, Abhijeet Sahu, and Peter Graf. "Buildingsbench: A large-scale dataset of 900k buildings and benchmark for short-term load forecasting." Advances in Neural Information Processing Systems 36 (2023): 19823-19857.

**Relation To Prior Work:**

Prior work is clearly discussed.

**Summary And Contributions:**

This paper introduces the Building Timeseries dataset (BTS). BTS offers access to a comprehensive sensor timeseries from three buildings over a three year period. Few previous datasets offer such comprehensive real-world building data, and the most closely related dataset only has data for a single building. BTS is intended to be used for developing building-agnostic machine learning (ML) techniques for building analytics. Two challenges, timeseries ontology classification and zero-shot forecasting, are demonstrated in the paper. Access to (some of the) data, and code, are made publicly available.

---

> ### Author Rebuttal · Authors · 2024-08-17
>
> We thank the reviewer for the thorough review of our paper and detailed feedback.
>
> > While an increase from 1 to 3 buildings is an improvement over the previous related dataset, I think it is still too few buildings to be impactful for transfer learning research and zero-shot generalization research.
>
> We agree with the reviewer and we endeavour to add more buildings into the dataset. This dataset release is part of NSW DIEF project (https://research.csiro.au/dch/projects/nsw-dief/), and we expect hundreds of buildings to be onboarded to our DCH platform (https://research.csiro.au/dch/). We hope that a fraction of these buildings would agree to have their data be made publicly available. Additionally, we are in the process of integrating Mortar (https://mortardata.org/) into our dataset, which currently have limited availability and accessibility.
>
> > Proofreading and revising. This includes making the presentation more concise and clear by improving the description of key dataset details currently relegated to the appendix.
>
> We appreciate the reviewer's feedback and agree that the main text should prioritize the details of the datasets and the classification benchmark. To accommodate this, we will relocate the zero-shot forecasting benchmark to the appendix. Furthermore, we will conduct a thorough proofreading. We thank the reviewer for these suggestions to enhance the paper's clarity.
>
> > While I really like the multimodal timeseries and knowledge graph aspect of the dataset, I felt that the KG aspect of the dataset was not introduced clearly. I don't think "ontology" is used properly throughout the paper (e.g., instead of 240 "ontologies", did the authors mean "entities"?). A formal, mathematical description of the KG and how it is aligned with the timeseries would help inform future multimodal research endeavors.
>
> We agree with the reviewer regarding the inconsistent use of some terms such as "ontology" and "classes". To improve clarity, we will conduct a thorough edit of the paper to ensure a consistent use of these terms. In particular, we will use the term "ontology" to describe the standardised hierarchical structure and relationships between concepts within the knowledge graph, and the term "class" to describe specific categories or types of entities within the ontology.
>
> We agree with the reviewer that a formal mathematical description of the knowledge graph is needed. We present the description as follows and we will integrate it into the paper:
>
> A building contains many different entities, such as equipment in various locations, and these entities are interconnected. A structure that captures this information is called a "building semantic model" and can be interpreted as a knowledge graph. The mathematical formalisation of the "building semantic model" is a directed acyclic graph $\mathcal{G}=(V,P,E)$ where:
>
> * Vertices (V): Each vertex $v \in V$ represents an entity within the building. This could be a physical location (e.g., a room or a zone served by a single HVAC subsystem), a piece of equipment (e.g. an air temperature sensor or a fan coil unit), or a reference to a time series in the form of a unique key. The actual time series data is typically stored in a separate database.
>
> * Edges (E): Each edge $e=(u,p,v) \in E$ represents a predicate $p$ between two vertices $u$ and $v$.
>
> * Predicate (P): Each edge $e$ is associated with a predicate $p \in P$ that specifies the type of relationship it represents (e.g., hasPart, has Location, or isPointOf).
>
> > To make the benchmark section more rigorous, more baselines can be included (in addition to hyperparameter tuning, which the authors acknowledge in Section 5). For example, the rule-based persistence baselines in [1] are a common and powerful baseline to include for short-term zero-shot forecasting on buildings.
>
> We would like to thank the reviewer for pointing out the unexpected F1 score in Table 3. Based on this feedback, we investigated our code, identified and fixed the underlying issue, and updated the GitHub repository with the corrected code https://github.com/cruiseresearchgroup/DIEF_BTS/commit/15e198b33a9e75e7732945589c25c754e1055ccb.
>
> Subsequently, we re-ran the experiments. As suggested, we included an additional set where we performed a hyperparameter tuning on the best-performing model, Transformer. The visualisation of the hyper parameter tuning and the updated results from all of the models are attached. Moreover, we agree with the reviewer that adding persistence baselines to the zero-shot benchmark is useful, the results are also attached. The paper will be revised to include these.
>
> > Discussing more potential uses-cases for the dataset would be helpful.
>
> To enhance the paper and address the reviewer's feedback, we propose incorporating the following potential use cases for the dataset at the end of Section 2.2:
>
> * **Control Challenges:** Develop and benchmark control algorithms to maximize occupant comfort, indoor environmental quality, and energy efficiency while minimizing carbon emissions and operating costs.
> * **Generative AI for Privacy-Preserving Data Sharing:** Explore the use of generative AI to create synthetic building timeseries data, enabling building owners to contribute data for research while safeguarding sensitive information.
> * **LLM Integration for Natural Language Interaction:** Investigate methods to integrate large language models (LLMs) with building timeseries data, allowing various stakeholders such as building operators to interact with and query the data using natural language.
> * **Redeployability:** By using a standardised ontology to describe the building, and linking timeseries data to the building model, applications (e.g. measurement and verification, chiller scheduling, occupant comfort) can be written to deploy against a fleet of buildings without a deep understanding of the building topology, such as those provided within this dataset.

---

> > ### Comment · Reviewer_eVsz · 2024-08-23
> >
> > Dear authors, thank you for taking the time to respond to my review. I appreciate your responses. Addressing most of my feedback requires edits to the final camera ready version, and although the authors have promised to do so, they were not able to provide a revised version of the manuscript at this time. Nevertheless, I will raise my score from 5 to 6.

---

> > > ### Author Response · Authors · 2024-08-24
> > >
> > > Thank you for your feedback and for increasing the score. Unfortunately, we are unable to present a revision at this stage, as revisions are not permitted, according to the NeurIPS FAQ. We sincerely appreciate your detailed and constructive feedback. Rest assured, we will incorporate your suggestions into the camera-ready version, as your insights are valuable and critical in improving the quality of our paper.

---

> ### Author Response · Authors · 2024-08-21
>
> Dear Reviewer, we have taken all your comments very seriously, and have revised the paper based on your valuable comments. We sincerely hope to utilize the remaining time to engage in a productive dialogue with you. We would like to hear your thoughts given our hard work in preparing the rebuttal and __**additional experiments**__ (please check the attached pdf in the rebuttal). Thank you again for your kind attention.

---

### Official Review · Reviewer_1N9M · 2024-07-29

**Rating:** 7
**Confidence:** 4

**Review:**

The authors present a 3 year dataset from 3 buildings in Australia. Often, such dataset require a lot of work that is behind the scenes, like permissions and discussions with the facility managers. Thus, the dataset contribution is a valuable one. While the authors do not discuss the efforts towards standardisation, I feel that must have been substantial. The two benchmarks are also very valuable to the community. I can see this dataset being useful for variety of tasks. Overall, the dataset and paper are impressive.

**Strengths:**

- The dataset is clearly one of the bigger datasets for non-residential buildings covering a large number of heterogeneous end points.
- The benchmarks tasks are important and well established.
- The problem is an important one and the dataset would help the field grow.
- The paper is well written.

**Additional Feedback:**

No other feedback.

**Clarity:**

The paper is well written. However, I have at points in this review suggested if some of the text from the appendix can be brought into the main text. However, that will not be trivial given the space constraints.

**Correctness:**

The dataset instrumentation and collections details are a bit scant. I would like the authors to comment more on this in the rebuttal to assure us of the correctness of the collected data and labels. Perhaps, some small multi reviewer checks on a subset of points?

**Documentation:**

The documentation on Github looks reasonable. There is no explicit hosting plan mentioned. But, I suppose with Github, this is not needed. The repo is MIT licensed. I believe the dataset and benchmarks looks reproducible.

**Ethics:**

The dataset provides various endpoints and is anonymised. I think more details on the anonymisation process would give more assurance on this aspect. However, there does not seem to be any significant ethical concern.

**Limitations:**

The paper does not really have any strong negative social impacts.

**Opportunities For Improvement:**

- The ontology problem would have to be clarified in more detail in the main text. For people outside the buildings energy community, this problem is not clear. Moving it from the appendix to the main text would be useful.
- Is the focus on non-residential buildings only? I suppose the residential buildings do not have the scale of the points mentioned in this study. Perhaps, focusing on them and motivating accordingly might be helpful.
- It would be good to give some details on what all instrumentation existed earlier, and what all instrumentation the authors had to do. Similarly, how much time and effort it took to map to the Brick schema.
- Perhaps, the authors have good reasons to stick to Brick schema. But, the audience would benefit from knowing why they used this schema.

**Relation To Prior Work:**

I feel the paper discussed the prior work well. Perhaps, some of the text from the appendix can be brought into the main paper. e.g. some thing like Table 5 from appendix might be useful to contextualise this dataset.

**Summary And Contributions:**

Buildings contribute significantly to overall energy. This paper presents a large 3 year dataset from 3 buildings in Australia containing a large number of instrumented points. The authors standardise the dataset and perform two benchmark tasks.

---

> ### Author Rebuttal · Authors · 2024-08-17
>
> We thank the reviewer for their insightful and constructive feedback.
>
> > The ontology problem would have to be clarified in more detail in the main text. For people outside the buildings energy community, this problem is not clear. Moving it from the appendix to the main text would be useful.
>
> We acknowledge the importance of clarifying problem definition of the classification task for readers outside the building energy community. We will move the relevant discussion from the appendix to the main text. Additionally, we will also include the attached figure to visualise the task.
>
> > Is the focus on non-residential buildings only? I suppose the residential buildings do not have the scale of the points mentioned in this study. Perhaps, focusing on them and motivating accordingly might be helpful.
>
> Non-residential buildings offer a diversity of equipment and building topology, commercial scale building loads, along with unique occupancy behaviours during times of peak renewable generation. For these reasons, the study has focussed on non-residential buildings, however investigation into residential buildings is identified for further study.
>
> > It would be good to give some details on what all instrumentation existed earlier, and what all instrumentation the authors had to do. Similarly, how much time and effort it took to map to the Brick schema.
>
> All instrumentation was conducted prior to the study, and as such no equipment installation or hardware setup was required by the authors. The work integrates with CSIRO’s Data Clearing House platform (https://research.csiro.au/dch/) which provides digital infrastructure to house building data, as well as to generate semantic models (e.g. via the BRICK schema) to describe the topology and instrumentation installed within the building. In terms of effort to map to the BRICK schema, once sufficient details about the building are compiled, then typically expert engineers requires at least one to two days of per building to generate a full semantic building model. We thank the reviewer for pointing out that this is an important info that is missing from the manuscript and we will add it.
>
> > Perhaps, the authors have good reasons to stick to Brick schema. But, the audience would benefit from knowing why they used this schema.
>
> We appreciate the reviewer's insightful comment regarding our choice to use the Brick schema. We have previously conducted a systematic evaluation of existing ontologies suitable for our research context, and the findings were published in [1]. The results of this evaluation informed our decision to adopt the Brick schema, as it was found to be the most appropriate for our specific needs. We will incorporate this information into the final manuscript, and we thank the reviewer for the valuable feedback, improving the clarity and completeness of our manuscript.
>
> [1] Zhangcheng Qiang, Stuart Hands, Kerry Taylor, Subbu Sethuvenkatraman, Daniel Hugo, Pouya Ghiasnezhad Omran, Madhawa Perera, Armin Haller, A systematic comparison and evaluation of building ontologies for deploying data-driven analytics in smart buildings, Energy and Buildings, Volume 292, 2023, 113054, ISSN 0378-7788, https://doi.org/10.1016/j.enbuild.2023.113054.

---

> > ### Comment · Reviewer_1N9M · 2024-08-28
> > **Thank you**
> >
> > Thank you for your detailed comments. I am happy with these. The score works in integers; I would have liked to increase it to something like 7.25 or 7.4, but there is no option.
> >
> > Best wishes to the authors.

---

> ### Author Response · Authors · 2024-08-21
>
> Dear Reviewer, we sincerely appreciate your encouraging review of our paper. WWe have addressed your questions in the rebuttal and found them very helpful in improving the clarity of our manuscript. Given the minor nature of the change, we are hopeful that our response meets with your continued approval. We also value the opportunity to engage in further discussion with you and hope to utilize the remaining time for a productive dialogue. Also note that, based on suggestions from other reviewers, we have incorporated __**additional experiments**__ into our work. For more details, we kindly invite you to check the general rebuttal or the specific thread dedicated to reviewer 1N9M. Thank you again for your kind attention and valuable feedback.

---

### Author Rebuttal · Authors · 2024-08-18

Dear Reviewers,

We appreciate your encouraging and constructive feedback, which has helped us refine and enhance our paper. We are pleased to inform you that we have addressed your requests with additional experiment results.

The __**attached**__ supplementary materials include the following:

1. Visualization of the hyperparameter tuning of the Transformer model.

2. New benchmark results on the timeseries multi-label classification task.

3. Benchmark results on the zero-shot forecasting task, including the persistence baselines.

We believe these additions enhance the overall quality of our benchmarks. We once again express our gratitude for your valuable insights and guidance.

---

### Decision · Program_Chairs · 2024-09-26

**Decision:**

Accept (Poster)

**Comment:**

The paper describes a new datasets with building time series data in Australia, containing lots of building analytics and features. Compared to previous similar datasets, this is richer and bigger, making it a valuable resource for the community. All reviewers are (very) positive, thus, the paper should be accepted.